# The Adaptation of the Wechsler Intelligence Scale for Children—5th Edition (WISC-V) for Indonesia: A Pilot Study

**DOI:** 10.3390/jintelligence13070076

**Published:** 2025-06-24

**Authors:** Whisnu Yudiana, Marc P. H. Hendriks, Christiany Suwartono, Shally Novita, Fitri Ariyanti Abidin, Roy P. C. Kessels

**Affiliations:** 1Donders Institute for Brain, Cognition and Behaviour, Radboud University, 6525 GD Nijmegen, The Netherlands; whisnu.yudiana@donders.ru.nl (W.Y.); marc.hendriks2@donders.ru.nl (M.P.H.H.); 2Department of Psychology, Faculty of Psychology, Universitas Padjadjaran, Jatinangor 45363, West Java, Indonesia; s.novita@unpad.ac.id (S.N.); fitri.ariyanti.abidin@unpad.ac.id (F.A.A.); 3Center for Psychological Innovation and Research, Faculty of Psychology, Universitas Padjadjaran, Jatinangor 45363, West Java, Indonesia; 4Centre for Psychometrics Study, Faculty of Psychology, Universitas Padjadjaran, Jatinangor 45363, West Java, Indonesia; 5Faculty of Psychology, Atma Jaya Catholic University of Indonesia, Jakarta 12930, Indonesia; christiany.suwartono@atmajaya.ac.id; 6Academic Centre for Epileptology, Kempenhaeghe, 5591 VE Heeze, The Netherlands; 7Center for Relationship, Family Life and Parenting Studies, Faculty of Psychology, Universitas Padjadjaran, Jatinangor 45363, West Java, Indonesia; 8Vincent van Gogh Institute for Psychiatry, 5803 DN Venray, The Netherlands; 9Radboudumc Alzheimer Center, Radboud University Medical Center, 6500 HB Nijmegen, The Netherlands

**Keywords:** psychological assessment, intelligence testing, WISC-V, adaptation, reliability

## Abstract

The Wechsler Intelligence Scale for Children (WISC) is a widely used instrument for assessing cognitive abilities in children. While the latest fifth edition (WISC-V) has been adapted in various countries, Indonesia still relies on the outdated first edition, a practice that raises substantial concerns about the validity of diagnoses, outdated norms, and cultural bias. This study aimed to (1) adapt the WISC-V to the Indonesian linguistic and cultural context (WISC-V-ID), (2) evaluate its psychometric properties in a pilot study with an Indonesian sample, (3) reorder the item sequence of the subtests according to the empirical item difficulty observed in Indonesian children’s responses, and (4) evaluate the factor structure of the WISC-V-ID using confirmatory factor analysis. The adaptation study involved a systematic translation procedure, followed by psychometric evaluation with respect to gender, age groups, and ethnicity, using a sample of 221 Indonesian children aged 6 to 16 years. The WISC-V-ID demonstrated good internal consistency. Analysis of item difficulty revealed discrepancies in item ordering compared to the original WISC-V, suggesting a need for item reordering in future studies. In addition, the second-order five-factor model, based on confirmatory factor analysis, indicated that the data did not adequately fit the model, stressing the need for further investigation. Overall, the WISC-V-ID appears to be a reliable measure of intelligence for Indonesian children, though a comprehensive norming study is necessary for full validation.

## 1. Introduction

Intelligence is a core cognitive concept in neuropsychology, typically measured by standardized tests. Some researchers view intelligence as a hierarchically structured construct composed of several cognitive processes and domains, such as verbal comprehension, fluid reasoning, visual-spatial ability, working memory, and information processing speed ([12]; [13]; [58]; [69]). Others, however, emphasize the general intelligence factor (g factor) as a central component in the structure of cognitive abilities ([25]; [54]). Intelligence plays a crucial role in various domains of human life, including planning and problem-solving in daily activities, education, professional careers, and clinical practice ([51]). Specifically in children, intelligence is essential for characterizing and classifying individuals based on their cognitive strengths and weaknesses ([12]), identifying learning problems and disabilities ([12]; [37]), and determining the most effective teaching and instructional methods in the classroom ([42]). Clinicians such as neuropsychologists, developmental psychologists, or clinical psychologists utilize intelligence assessments to evaluate the cognitive profiles of children, including those with intellectual disabilities ([3]; [8]), developmental disorders such as Attention Deficit Hyperactivity Disorder ([50]), or brain disorders such as epilepsy ([34]).

Intelligence tests differ in content, format, and underlying theoretical frameworks. Their primary aim is to assess different intellectual domains, which may vary among children, even in those with similar general cognitive abilities ([5]). The Wechsler Intelligence Scales for Children (WISC) are among the most widely used tests for measuring children’s intellectual abilities worldwide ([2]; [40]; [63]). The first version of the WISC was published in the USA in 1949 ([67]). Since then, the test has undergone numerous refinements and modifications, both in terms of items, subtests, and norms, in response to updated conditions and theoretical developments, particularly driven by the rapid growth of research in cognitive neuroscience and functional brain imaging ([29]; [40]). These modifications were also motivated by the Flynn effect, which refers to substantial IQ gains from one generation to another within the twentieth century ([16]).

The most recent version of the Wechsler Intelligence Scales for Children, the fifth edition (WISC-V, [69]), was initially published in the USA in 2014. The test comprises 21 subtests, including 10 primary subtests, 6 secondary subtests, and 5 supplemental subtests. Since its release, the WISC-V has been translated and adapted worldwide. In most countries, such as Canada ([70]), the United Kingdom ([74]), Australia and New Zealand ([73]), and Taiwan ([76]), the WISC-V includes 16 subtests, primary and secondary subtests combined. In other countries, such as Spain ([71]), France ([72]), Germany ([75]), and Chile ([56]), the 15-subtest version was published, excluding the secondary subtest Picture Concepts. In the Netherlands ([21]), a 14-subtest version was published, excluding the secondary subtests, Information and Comprehension. These adaptations and studies reflect the generalizability of the intelligence structure proposed in the WISC ([63]; [78]). Despite its potential applications in assessing various aspects of children’s cognitive abilities, reports on the adaptation of the WISC-V in Southeast Asian countries, including Indonesia, remain relatively scarce. To date, the only Wechsler test that has been adapted and standardized for use in the Indonesian context is the Indonesian version of the Wechsler Adult Intelligence Scale-IV (WAIS-IV-ID) ([60]), which also includes the development of a short-form version ([61]).

The adaptation of the WAIS-IV-ID followed the guidelines for test adaptation set by the International Test Commission ([22]; [24]). Currently, the process is in its final stages before the test can be used in Indonesia. In contrast, intelligence measures for children in Indonesia are, to date, limited to unauthorized translations of the first edition of the WISC ([67]) and the WISC-R ([68]), conducted at the time without formal permission from the test publisher. Furthermore, the widespread use of the WISC in Indonesia—such as for clinical and educational practice, including in the curricula of psychology training programs at most universities—has not been accompanied by a well-documented written manual and, more importantly, has never been standardized and normed in the Indonesian population.

The use of the first edition of the WISC without standardization or normative data for the Indonesian population raises concerns about its validity in assessing intelligence in the Indonesian context. One study suggests that relying on non-representative norms may lead to misclassification and an invalid understanding of children’s cognitive abilities ([35]). For example, a child may be classified as below average due to the use of inappropriate, culturally biased norms. Other research has emphasized the importance of adopting local norms to ensure accurate interpretation of cognitive test results ([7]). In addition, it is difficult to argue that the current version of the WISC, which was developed in 1949 and 1974 without an authorized translation, can be used without any concern in the 21st century. Therefore, the current versions of the WISC used in Indonesia carry the risk of measurement errors, misdiagnosis of children’s intellectual abilities, and wrong decisions in indicating interventions (i.e., treatment programs for learning difficulties). Therefore, a formal, authorized adaptation of the Wechsler Intelligence Scale for Children (i.e., the WISC-V) for Indonesia is necessary and urgent. The purposes of this article are to (1) describe the adaptation processes of the WISC-V for use in the Indonesian context, (2) evaluate the psychometric properties of the WISC-V-ID using an Indonesian sample, (3) reorder the item sequence of the subtests according to the empirical item difficulty observed in Indonesian children’s responses, and (4) evaluate the factor structure of the WISC-V-ID using confirmatory factor analysis.

## 2. Materials and Methods

### 2.1. Participants

This pilot study was conducted in the province of West Java, Indonesia, the most populous province in Indonesia, comprising 18% of Indonesia’s population in 2020 ([4]). The study included 221 children aged 6 to 16 (*M* = 11.40, *SD* = 3.16), of whom 51% (113) were girls. Of these, 4.98% were in kindergarten, 49.32% were in primary school (years 1 to 6), 27.15% attended middle schools (years 7 to 9), and 18.55% were in high school (years 10 to 11). Regarding ethnicity, 42.08% were Sundanese, 24.43% Batak, 18.10% Javanese, 9.05% were Chinese, and the remainder belonged to other ethnicities. All participants were native Indonesian speakers without apparent physical or intellectual disabilities that could interfere with test administration.

### 2.2. Instrument

In the present study, the fifth version of the Wechsler Intelligence Scale for Children-5th edition was used to adapt for Indonesian speakers (referred to as the WISC-V-ID). Similar to the original version, the test comprises 10 primary subtests, namely Block Design (BD), Similarities (SI), Matrix Reasoning (MR), Digit Span (DS), Coding (CD), Vocabulary (VC), Figure Weights (FW), Visual Puzzles (VP), Picture Span (PS), and Symbol Search (SS). Originally, these primary subtests are used to estimate five Primary Index scores: Fluid Reasoning, Verbal Comprehension, Visual Spatial, Working Memory, and Processing Speed. The test also includes six secondary subtests: Information (IN), Picture Concepts (PC), Letter-Number Sequencing (LN), Cancellation (CA), Comprehension (CO), and Arithmetic (AR).

### 2.3. Adaptation Process of the WISC-V-ID

The translation and adaptation of WISC-V-ID followed the guidelines of the International Test Commission ([22]; [24]) and the International Neuropsychological Society ([44]) and involved the following phases:

#### 2.3.1. Permission for Translation and Adaptation

Permission was obtained from the intellectual property rights holder, Pearson Education South Asia Pte Ltd, Sydney, Australia, through a Statement of Work (SOW) agreement with Universitas Padjadjaran, Indonesia, dated 27 March 2023.

#### 2.3.2. Expert Review

Expert consultations identified the specific components requiring translation and adaptation, which include two aspects. First, the Indonesian translation of instructions, administration procedures, and scoring guidelines. Second, item translations and adaptation for selected subtests of Verbal Comprehension (SI, VO, IN, and CO) and Fluid Reasoning (AR).

#### 2.3.3. Forward and Backward Translation

Two bilingual translation teams conducted forward and backward translations, ensuring cultural and linguistic equivalence. Each team consisted of three members fluent in English and Bahasa Indonesia, two of whom had a PhD in psychology and one of whom was a clinical child psychologist. All the translators also had experience in psychometrics and had previously conducted test adaptations. The following steps were taken to ensure the quality of both forward and backward translations:

The first step involved the forward translation of the administration procedures and scoring guidelines for 16 subtests, as well as all items from five subtests (SI, VO, IF, CO, and AR). Initially, all translators worked independently to review, translate, and modify the items to ensure alignment with Indonesian culture and local context while maintaining equivalence with the original items. Where specific nomenclature or terminology was unfamiliar in the Indonesian context (e.g., the word “Winter”), it was replaced with a more familiar nomenclature (e.g., “Rain”). The translation coordinator then compiled all forward translations into a single document. Any discrepancies in terminology or items were discussed to reach a consensus, with the involvement of an Indonesian language expert to ensure an accurate and equivalent adaptation process.

In step 2, the backward translation was conducted by three independent translators. All items with discrepancies in meaning between the Indonesian and English versions were discussed to arrive at a single translation result. Next, an independent committee reviewed and discussed the forward and backward translation results and the original items to achieve the most accurate adaptation.

Across the five subtests of the original Verbal Comprehension Index (VCI), the percentage of items undergoing adaptation for the Indonesian context varied from 3% to 31%. The VO subtest underwent the most significant changes. Some items in the original version assessed C1–C2 English proficiency, assuming advanced language skills. However, due to the difficulty of finding Indonesian equivalents of similar levels of difficulty, these items were replaced with more commonly used words with simpler meanings. Additionally, several example answers were added to key questions related to “remedy”, as Indonesian children were likely to interpret the term in the context of improving a low exam grade rather than its original meaning related to “cure” or “medicine”. Adjustments were also made to other VCI subtests. In the SI subtest, items referencing unfamiliar concepts were modified. For example, besides an item about US seasons being adapted to reflect Indonesia’s two-season climate, the item concerning sibling relationships (e.g., “brother and sister”) was also adapted because the Indonesian language does not specify gender. In addition, an item referring to a specific time-measuring instrument unfamiliar to Indonesian children was replaced with a more commonly recognized Indonesian timekeeping tool. The IN and CO subtests were similarly modified. Geographic locations unfamiliar to Indonesian children were replaced with recognizable cities in the IN subtest. In the CO subtest, an item related to the children’s reasons for traveling was replaced with an activity more familiar to Indonesian school children. Finally, the pronouns in the AR subtest were adjusted to sound more natural in Indonesian, as the originals were based on US child names.

Step 3 involved a small-scale field test of the first version of the Indonesian WISC-V (WISC-V-ID) with five Indonesian children as a target population. The administration was administered by members of the adaptation team and observed by two junior psychologists acting as data collectors and assessors. The purpose of the field test was to assess how feasible the adapted administration procedures and items were when applied in the Indonesian context. During testing, administrators provided additional information, simplified instructions, and made modifications as needed when children experienced difficulties in understanding. After testing the children, the adaptation team and assessors reviewed both the testing experience and the children’s feedback, then adjustments were made as necessary to improve clarity and usability. All adjustments were documented to finalize the administration procedures, which were then documented as the final translated version of the WISC-V-ID.

### 2.4. Procedure

The data collection procedures were registered and approved by the Research Ethics Committee of Universitas Padjadjaran. Permission was obtained from the school principals and teachers, written consent was obtained from one or both parents, and assent was also obtained from the participating children. Data collection was conducted individually by Master’s students or graduates with a Bachelor’s degree in Psychology who had undergone a two-day training on the administration and scoring of the adapted WISC-V-ID. The training included role-playing exercises before examiners commenced data collection with children. To ensure the standardization and quality of the test administration, each examiner was required to submit recordings of their first two data collection sessions and await feedback before proceeding with further testing. Any identified errors were reviewed and discussed with the examiners for improvement, as outlined in the administration manual ([69]).

The test was administered following the guidelines outlined in the administration manual ([69]). Each child was tested individually, beginning with 10 primary subtests. After a brief break, the test continued with six secondary subtests. In this pilot study, the item order mirrored the original US version, and discontinue rules were not applied; all items were administered to each child. On average, the test administration ranged from 2 to 3 h, depending on the child’s pace. After completion, the children received a voucher or lunch box and snacks valued at Rp50,000 (approximately $3.00 or €2.80).

### 2.5. Analyses

The analyses were conducted to evaluate the psychometric properties of the WISC-V-ID, focusing on item discrimination, difficulty, and test reliability. Thirteen of sixteen subtests of the WISC-V-ID were analyzed, while CD, SS, and CA were excluded from the analysis since they rely on time constraints rather than individual correct answers.

The majority of subtests in the WISC-V-ID consist of dichotomous items; however, five out of the sixteen subtests (i.e., BD, SI, PS, VO, and CO) contain polytomous items. Item discrimination measures the ability of the items to differentiate between high- and low-achieving children ([9]). In this study, item discrimination was estimated using the item-total correlation for each subtest. Correlations were primarily based on Pearson product-moment correlation for polytomous items, whereas biserial point correlations were used for dichotomous items. Coefficients were expected to be positive, and items with negative coefficients were subject to revision or removal ([9]).

Item difficulty (*p*) was typically estimated using the average item score. For dichotomous items, this index reflects the proportion of children who answered the item correctly, with values ranging from 0 to 1 (where 1 indicates an easy item) ([9]). This analysis is crucial for reordering the WISC-V-ID items based on their difficulty. Reordering was necessary because, in the WISC-V and previous versions, participants did not complete all subtest items but instead progressed based on ability, with discontinue rules in place to stop testing after consecutive incorrect answers ([69]).

Cronbach’s α coefficients were calculated to assess internal consistency, reflecting the extent to which items within each subtest measure the same underlying construct. However, given the limitations of α in accurately estimating the reliability of multidimensional scales, which are common in neuropsychological assessments ([65]), McDonald’s ω was also computed in this study. McDonald’s ω was selected because it relies on fewer assumptions and is increasingly recommended for reporting internal consistency, particularly in scales that may not meet the strict assumptions required by α ([14]). In the US version, reliability coefficients were reported separately by age ([69]). However, due to constraints in the sample size and to ensure more homogeneous characteristics within each group, this pilot study only reported reliability estimates across three broader age groups: 6–9 years, 10–12 years, and 13–16 years. Each group comprised approximately 66 to 88 participants. Since CD, SS, and CA assess processing speed based on all items taken together, reliability estimates could not be computed for these subtests. Several recommendations have been proposed for interpreting the internal consistency of psychological tests. A commonly cited minimum threshold is 0.70, while values between 0.80 and 0.90 are often recommended when tests are used for individual-level decision-making ([9]; [28]). In clinical settings, high reliability (with a value of at least 0.95) is critical due to the potential impact of test results on an individual’s diagnosis ([9]; [28]; [46]). To aid interpretation, the standard error of measurement was also reported. 

As mentioned in the Section 2.4, the test was administered using the original US version without applying the discontinue rules. However, to further examine the quality of the test, this study also calculated internal consistency estimates after reordering the items based on the observed item difficulty in the sample and applying the discontinue rules as recommended in the test manual ([69]). This approach was used to conduct a comparison of reliability estimates under different scoring conditions, providing additional evidence regarding the internal consistency of the WISC-V-ID. To evaluate the effects of item ordering and the application of discontinue rules, a 2 × 2 factorial analysis was conducted. In addition to the two primary conditions, reliability estimates were also calculated for two supplementary conditions: the original item order with discontinue rules applied, and the reordered items without rules being applied (See Appendix A, Table A2 and Table A3).

This pilot study involved conducting confirmatory factor analysis (CFA) to provide evidence of the WISC-ID’s validity evidence based on its internal structure. Two models were evaluated. The first model, based on the WISC-V test manual ([69]), specifies a second-order five-factor structure representing the following broad cognitive abilities: (1) Verbal Comprehension (Gc), measured by VO, SI, IN, and CO; (2) Visual Spatial (Gv), measured by BD and VP; (3) Fluid Reasoning (Gf), measured by MR, FW, and PC; (4) Working Memory (Gwm), measured by DS, PS, LN, and AR; (5) Processing Speed (Gs), measured by CD, SS, and CA. These five first-order factors were modeled under a higher-order general intelligence factor (g). The second model was a modified version informed by prior findings in the WISC-V manual, which indicated that the AR subtest may cross-load onto both Gf and Gc ([69]; [77]). CFA was conducted for each age group, as well as for the full sample of 6–16-year-olds evaluated on the original version.

Multiple indices were employed to assess model fit ([18]; [30]). The chi-square (χ^2^), root mean square error of approximation (RMSEA), and standardized root mean square residual (SRMR) were used to evaluate the absolute fit of the model. RMSEA and SRMR values of 0.06 or less are indicative of a good model fit ([23]), whereas an RMSEA value of 0.10 or higher is typically considered evidence of a poor fit ([36]). However, the chi-square statistic is known to be highly sensitive to large sample sizes and may overestimate model misfits ([30]). Relative model fit was assessed using the comparative fit index (CFI) and the Tucker–Lewis index (TLI). Values of 0.95 or higher are generally considered to reflect a good fit to the data ([23]). Based on the model, internal consistency estimates of reliability were calculated. In Addition, correlations were calculated between raw subtest scores and the children’s ages as a measure of the validity, with positive correlations expected.

All analyses were conducted using R statistical software ([53]). Descriptive statistics and reliability analyses were performed using the *psych* package ([55]). The 2 × 2 factorial analysis was performed using the *factorial2x2* package ([33]), while confirmatory factor analyses were conducted using the *lavaan* ([57]) and the *semTools* packages ([26]).

## 3. Results

Table 1 summarizes the descriptive statistics of each of the WISC-V-ID subtests for the total sample (*N* = 221). When comparing the maximum test scores achievable by children and the mean correct scores, it was evident that, on average, nearly half of the items were answered correctly by the children. The FW (*M* = 21.76, *SD* = 4.80), MR (*M* = 18.77, *SD* = 4.20), and PS (*M* = 28.43, *SD* = 8.04) subtests were the three highest-scoring subtests, with an average of around 56–64% of the items answered correctly by children. Conversely, the CD *(M* = 47.22, *SD* = 18.29), CO (*M* = 15.63, *SD* = 5.90), and IN (*M* = 16.63, *SD* = 4.74) subtests had the three lowest scores, with an average of around 40–44% of the items answered correctly by children.

Table 2 summarizes the item discrimination and difficulty indexes for each subtest of the WISC-V-ID. The correlation between items and total scores varied in strength across the subscales, with the average item discrimination index ranging from 0.34 to 0.47. For dichotomous subtests, the average item difficulty index ranged from 0.49 to 0.64. For polytomous subtests scored on a 0–2 scale (SI, VO, PS, and CO), the average item difficulty ranged from 0.87 to 1.09. In addition, the average item difficulty for the BD subtest, which is scored on a 0–7 scale, was 2.17. However, a total of eight items exhibited negative item discrimination indexes, that is, four items in the MR subtest, two in the FW subtest, one in the PC subtest, and one in the AR subtest. Seven out of these eight items had a difficulty level categorized as hard (*p* < 0.30), while only one item from the AR subtest was categorized as easy (*p* = 0.80) (this criterion was developed according to [9] ([9])). The item difficulty analysis was performed to reorder the items in the WISC-V-ID according to their difficulty levels.

Internal consistency was estimated using Cronbach’s α and McDonald’s ω, based on the original item order, with all items administered and no discontinue rules applied. Estimates were calculated for the overall sample and separately for each age group, as detailed in Appendix A, Table A1. Across the full sample, Cronbach’s α values ranged from 0.76 to 0.89, and McDonald’s ω values ranged from 0.73 to 0.90. The MR subtest consistently demonstrated the lowest internal consistency across both indices and age groups. For children aged 6–9, reliability estimates were generally high, with coefficients ranging from 0.74 to 0.90 across most subtests. In the 10–12 age group, the MR and CO subtests yielded values below the commonly accepted threshold of 0.70, indicating a need for further analysis. Additionally, the FW subtest yielded a notably low value of ω = 0.42. Among children aged 13–16, three subtests (MR, PC, and LN) produced α and ω coefficients ranging from 0.61 to 0.63 for both, which fall below the generally accepted standard for internal consistency.

Table 3 shows the internal consistency of the subtests after the items were reordered and the discontinue rules applied. Internal consistency estimates based on Cronbach’s α and McDonald’s ω were high for the overall age group, ranging from 0.81 to 0.94. Across all age groups, Cronbach’s α coefficients met acceptable standards, ranging from 0.72 to 0.91. The lowest value was observed for the CO subtest in the 10–12 age group. Similar results were found using McDonald’s ω, which ranged from 0.70 to 0.93. However, the LN subtest in the 13–16 age group warrants attention, as the ω value was 0.47, falling below the commonly accepted reliability threshold.

A 2 × 2 factorial analysis examined the effects of item ordering (Original vs. Reordered) and discontinue rule application (Without vs. With) on Cronbach’s α and mean scores. The analysis showed no effect of item ordering on Cronbach’s α (*F*(1, 12) = 0.005, *p* = 0.94), but applying discontinue rules significantly increased Cronbach’s α (*F*(1, 12) = 20.53, *p* < 0.001; *M* = 0.88 [*SD* = 0.03] vs. 0.84 [*SD* = 0.04]). Meanwhile, a follow-up analysis of mean subtest scores found that the application of the discontinue rules led to lower scores (*F*(1, 12) = 35.38, *p* < 0.001; *M* = 19.38 vs. 20.29). Interestingly, the mean score for the group with reordered items and applied to discontinue rules was slightly higher than those for the group with the original item ordering and applied to discontinue rules (*M* = 19.38 vs. 19.20), though this difference was not statistically significant (*t*(12) = −1.00, *d* = −0.03).

Table 4 shows the goodness-of-fit indices for the confirmatory factor analyses of Model 1 and Model 2. Model 1 is a second-order five-factor model, while Model 2 retains the same structure but allows the AR subtest to cross-load onto both Gc and Gf. The results indicate that Model 1 did not meet the recommended fit criteria ([23]): RMSEA values exceeded the suggested cutoff point of 0.06 in all age groups and in the overall sample. In contrast, Model 2 showed a slight improvement in the fit, particularly for the overall age sample. This model demonstrated acceptable fit indices (CFI = 0.96, TLI = 0.95, SRMR = 0.04), although the RMSEA remained marginally above the recommended cutoff at 0.07. These findings suggest that the model fit improves when the AR subtest is permitted to load on both Gf and Gc, indicating that AR may tap into multiple cognitive domains. Notably, among the age groups, the 6–9 age group showed a marginal fit to the model (CFI = 0.95, TLI = 0.94, RMSEA = 0.07, SRMR = 0.06), while the other age groups exhibited lower fit indices.

Additionally, Figure 1 presents the standardized factor loadings for the five-factor model of the WISC-V in the overall sample. The loadings from the first-order factors to the observed variables were relatively high, ranging from 0.41 to 0.92. Notably, the AR subtest has a very low loading on Gf (0.10), compared to higher loadings on Gc (0.41) and Gwm (0.42). This pattern contrasts with findings from the US standardization sample, where AR had its lowest loading on Gc (0.16; see Wechsler 2014a). At the second-order level, the model produced an overestimated loading of Gf on the general factor (g), of 1.03, exceeding the theoretical maximum of 1.00, which may indicate an estimation issue or model misfit. This finding is consistent with similar results reported in prior research ([69]).

The reliability coefficients for the composite scores are presented in Table 5. For the overall sample, all composite scores showed reliability above 0.80, indicating good consistency to support individual-level decisions. The general intelligence factor (g) demonstrated excellent reliability at 0.96. However, examining the age groups more closely reveals that a few composite scores are cause for concern. For instance, the Gf composite had reliability values of 0.65 for children aged 10–12 and 0.69 for those aged 13–16. These reliability coefficients were below the commonly accepted threshold. The Gs composite score for the 13–16 age group was particularly low at 0.59.

Additionally, Table 6 demonstrates that age was significantly and positively correlated with all subtests of the WISC-V-ID. Overall, the correlations ranged from 0.44 to 0.76. Specifically, age showed a high correlation with all Verbal Comprehension subtests (SI, VC, IN, and CO) and some Processing Speed subtests (CD and CA).

## 4. Discussion

This study aimed to describe the cultural adaptation and translation process of the WISC-V and evaluate the psychometric properties of the WISC-V-ID in an Indonesian sample of 221 children, focusing on item difficulty, item discrimination, and reliability for each subtest. Overall, the results provided evidence of a successful adaptation process.

Specifically, this study aimed to reorder item difficulty based on the observed performance of Indonesian children aged 6 to 16. Findings indicated that item difficulty order in some of the WISC-V-ID subtests was similar to the US-English version ([69]), while others differed. This difference excludes the Processing Speed subtests (CD, SS, and CA) due to their time-based format, which did not require item-level adaptations. Following a previous study, the item reordering in the WISC-V-ID for this study was based on two factors: the conceptual framework of item sequencing in the US-English version and the observed item difficulty in this pilot study ([11]). For instance, the item order remained unchanged in the DS, PS, and LN subtests. These subtests assess working memory, where items are sequenced based on the increasing number of stimuli children need to remain active within working memory. Therefore, no order changes were made. Similarly, the BD subtest also maintained its original order. Data showed that BD’s item difficulty increased as the number of blocks used increased and the complexity of the stimuli went up. Additionally, minor adjustments were made to the item order in the MR, FW, VP, PC, and AR subtests, where on average, items were reordered only one or two positions up or down.

In contrast, subtests measuring the Verbal Comprehension Index underwent the most significant changes in item order, content, and difficulty. This was especially the case for the VC subtest. As previously mentioned during the adaptation process, approximately nine words originally classified as C1–C2 level in English were adapted to simpler meanings in Indonesian. This adaptation was necessary, in part, due to the limitations of Indonesian vocabulary in expressing complex concepts often found in global communication ([47]). For example, the item “frugal”, which relates to the concept of “prudently saving or sparing”, was translated into “*hemat*” in Indonesian. The word is a commonly used word that closely aligns with the meaning of “economical”, but there is no more complex term in Indonesian that fully captures the connotation of “frugal”. The item moved from position #26 to #13. The data revealed that most of these words originally classified as difficult became moderately difficult after adaptation. This suggests the chosen replacements were more commonly used by some of the children. In the SI subtest, items generally moved one to five positions up or down. The item asking about the similarity between “sour” and “salty” (likely easier for Indonesian children) moved from position #10 to #5. This suggests that the concept of taste might be easier for children to grasp because it relates to food, a concept that is more culturally important. Meanwhile, the item asking about the relation between a “shirt” and “shoes” moved from position #4 to #10, thus suggesting a higher difficulty level. This shift might be related to cultural clothing norms in Indonesia, where pants might be more common than shoes as paired clothing items for children.

The CO subtest required small changes in item order. However, an interesting finding was observed within this subtest. Two proverb items demonstrated differing patterns of difficulty. One item (#11, which addresses personal responsibility in learning) was more difficult for Indonesian children than for US children and was moved to Item #16. In contrast, Item #15 (which involves how to handle difficult situations) was found to be easier for Indonesian children and was moved to an earlier position, to Item #12. The accuracy in interpreting proverbs may be influenced by contextual understanding, abstract reasoning, metaphorical thinking ([45]), and familiarity with the proverb’s topic or phrasing in the Indonesian context ([39]). In this case, the question in Item #15 may appear to have been more culturally familiar to Indonesian children than the one in Item #11. In addition, the IN subtest also had some interesting changes in item order. One notable example is the item about an ancient limestone statue located in Egypt. This item moved from position #20 to #15, suggesting it became easier for Indonesian children. The possible reason for this is that familiarity with statues through stories or Islamic cultural references might have played a role.

Reporting of both Cronbach’s α and McDonald’s ω reliability coefficients has become increasingly important, particularly for newly developed tests including items with a wide range of factor loadings, both high and low ([14]; [20]). The analysis revealed that internal consistency estimates obtained using Cronbach’s α and McDonald’s ω were generally comparable across most subtests and age groups. Differences ranged from 0.01 to 0.04 points for the overall sample. Consistent with several previous studies, α tended to produce slightly lower values than ω. This reflects the conceptual distinction whereby α represents a lower bound of reliability, while generally ω provides a more accurate and typically higher estimate ([20]; [38]; [64]). It is important to understand that these estimates are critical because reliability directly influences the calculation of the standard error of measurement, which in turn affects the interpretation of results and decision-making processes ([64]).

Based on the overall sample, this study showed that the reliability of all WISC-V-ID subtests was satisfactory, with 11 out of 13 subtests exhibiting coefficients exceeding 0.80 (based on Cronbach’s α or McDonald’s ω). The DS and VC subtests showed the highest reliability estimates (ω = 0.90). However, the MR and PC subtests, which assess fluid reasoning using non-verbal stimuli (identifying relationships among objects), had slightly lower reliability coefficients, ranging from Cronbach’s α or McDonald’s ω = 0.73 to 0.79. Similar results were previously found when adapting the Wechsler Preschool and Primary Scale of Intelligence III (WPPSI-III) for a low-income setting, where BD, MR, and PC were the three subtests with the lowest reliability ([52]). The slightly lower reliability of these subtests may be explained by their relative difficulty compared to other subtests (see Table 2). In particular, several difficult items displayed negative item discrimination, further impacting reliability. In the current study, all items were administered to the children without discontinue rules. Unlike verbal tests, where participants can say “I don’t know,” the multiple-choice format (for MR and PC) lacked a “don’t know” option. This likely prompted children to guess on items they were unsure about, particularly on difficult items, negatively impacting the test’s reliability ([48]; [79]). Removing four items with negative discrimination indices from the MR subtest could increase reliability estimates from Cronbach’s α from 0.76 to 0.80 and McDonald’s ω from 0.73 to 0.77. In contrast, removing one item with a negative discrimination index from the PC subtest resulted in a slight increase in Cronbach’s α from 0.78 to 0.79, with no change observed in McDonald’s ω. However, an independent cross-validation sample is required to test these assumptions empirically, as conclusions cannot be drawn from the current analyses alone.

Internal consistency was also examined across three age groups: 6–9, 10–12, and 13–16 years. The results showed that the reliability values of nine out of thirteen subtests—including BD, SI, DS, FW, VP, PS, IN, and AR—had comparable reliability values to the overall Cronbach’s α coefficients, indicating consistent test quality across age groups. However, concerns were raised regarding the Cronbach’s α coefficients. An interesting finding was the substantial discrepancy whereby Cronbach’s α was notably higher than McDonald’s ω for the MR and FW subtests in the 10–12 age group. Several factors may explain this difference. Firstly, correlated measurement errors across items can inflate Cronbach’s α estimates ([14]; [20]). Another possible explanation is the presence of skewed item distributions and multifactorial measurement structures, which can also affect reliability estimates ([38]). However, this phenomenon was observed only in this specific age group and subtests, indicating the need for further examination.

Further analysis revealed a clear improvement in reliability after reordering the items according to difficulty and applying discontinue rules. These rules aim to improve test efficiency while maintaining subtest reliability. Research suggests that stopping the test after two or three incorrect responses is appropriate for fixed-format tests ([1]). Our study found similar results: applying the discontinue rules outlined in the manual after two or three incorrect responses ([69]), along with reordering the items from easiest to most difficult, resulted in a good to excellent reliability range for the test. Based on these reliability coefficients, items with negative discrimination indices in the MR and PC subtests were retained, since applying discontinue rules improved the overall quality of the subtests (see Table 3). Retaining these items also ensured that the test contained a sufficient number of items that met the ceilings and floors that were established in the US version ([41]). A factorial 2 × 2 analysis revealed that the application of discontinue rules significantly increased the reliability of the WISC-V-ID. Additionally, item reordering slightly increased the mean reliability compared to the original item order. These findings suggest that item reordering and the application of discontinue rules could be beneficial in future research. Applying discontinue rules is important for reducing testing time and participant fatigue. Any potential reduction in scores resulting from not administering all items can be addressed by developing norms based on the application of discontinue rules.

The manual reported that, based on confirmatory factor analysis (CFA), a secondary five-factor model incorporating 16 primary and secondary subtests was supported as the structural framework of the WISC-V. Furthermore, model fit indices improved significantly when the AR subtest loaded on both the Gc and Gf factors. This factor structure was also found to be consistent across different age groups ([69]). However, this pilot study could not fully replicate the secondary five-factor model in the data sample, for either the full sample or within age groups. This suggests that the WISC-V-ID may not fit well with the second-order five-factor model. The inadequacy of the second-order five-factor model has also been observed in various standardized samples, including the US version, and has been subject to debate by several researchers ([6]; [32]; [49]; [66]). Most studies have found that a four-factor model provides a better fit than the five-factor model ([6]) or a bifactor model with four group factors and one general factor ([32]; [49]; [66]). In these studies, Gf and Gv abilities were combined into a single factor.

Slightly different results were found when the AR subtest was loaded onto Gc and Gf: the model fit improved for the overall sample, but not for the individual age groups. Besides the questionable fit of the second-order five-factor structure, the poor model fit across age groups may be due to the small sample size in this study, which, per age group, was below the recommended minimum of 100 participants as suggested ([30]). This finding suggests that the model structure based on age groups still requires further examination. In this model, AR was more strongly associated with Gc and Gwm than with Gf, which differs from the US version ([69]). The composite reliabilities demonstrated good consistency, supporting their use for individual-level decisions.

To further support the validity of the WISC-V-ID, the correlation between subtest scores and age was examined. The results showed that the correlations ranged from moderate to high. The positive correlations between subtest scores and age likely reflect the rapid development of cognitive functions observed from early childhood to early adolescence ([43]; [62]). The subtests showing the strongest correlations with age were CD and CA, which measure processing speed. This positive correlation likely reflects the well-documented trend of processing speed abilities improving with age in children and young adults ([15]; [19]; [27]; [62]). As children age, their processing speed abilities continue to mature, leading to faster information processing, reaction times, improved matching skills, and better mental rotation abilities. Other subtests that correlated highly with age are SI, VC, IN, and CO, which measure Verbal Comprehension. The high correlation might be driven by a strong education effect on crystallized intelligence tests ([59]) and the strong language development from the age of 6 onwards that helps to define the word correctly ([10]; [31]). Moreover, the FW subtest, which measures fluid reasoning, was found to have a lower correlation with age compared to other subtests. Previous research suggests a non-linear pattern for fluid reasoning development, as opposed to crystallized intelligence, with rapid increases in childhood followed by a gradual rise in adolescence due to the uncertain development of how the ability to reason about relationships increases ([17]). This non-linear pattern might explain the FW subtest’s lower correlations with age in this study.

Overall, the results of this pilot study suggest that the WISC-V-ID could be feasible and reliable for use in Indonesia. However, several limitations in the current study suggest areas for future research. Firstly, this study primarily employed a classical test theory approach to examine test quality. Future research could benefit from applying item response theory (IRT) to provide more precise information on item characteristics, differential item functioning, and the effects of item reordering and the application of discontinue rules on test information—an approach that has been utilized in previous studies. Secondly, the pilot study involved a relatively small sample size. A larger sample is recommended to better identify the factor structure of the WISC-V-ID and to advance efforts to standardize and norm the test for a broader Indonesian population. Such studies should employ confirmatory factor analysis (CFA) based on the Cattell–Horn–Carroll (CHC) model (see [58]), exploratory structural equation modeling (ESEM), bifactor models, and multi-group invariance testing to address the cross-cultural generalizability of the WISC-V factor structure.

## Figures and Tables

**Figure 1 jintelligence-13-00076-f001:**
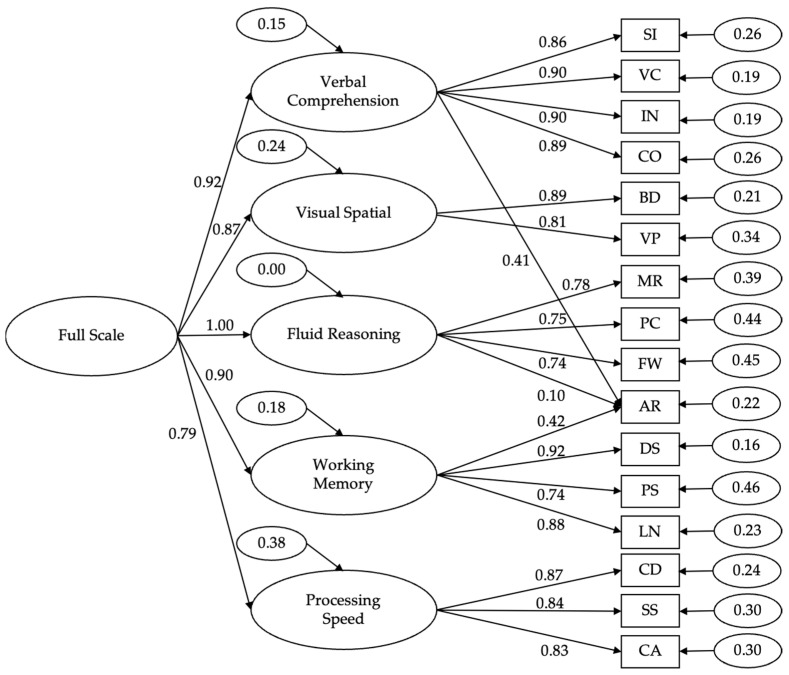
Five-factor second-order model for the WISC-V-ID. Note: Bold = internal consistency; regular = standard error of measurement. BD = Block Design; SI = Similarities; MR = Matrix Reasoning; DS = Digit Span; CD = Coding; VC = Vocabulary; FW = Figure Weights; VP = Visual Puzzles; PS = Picture Span; SS = Symbol Search; IN = Information; PC = Picture Concepts; LN = Letter–Number Sequencing; CA = Cancellation; CO = Comprehension; AR = Arithmetic.

**Table 1 jintelligence-13-00076-t001:** Descriptive data of the WISC-V-ID subtest scores.

Subtest	Maximum Score of Test	Correct Answer		
		Range	Mean	SD
Block Design (BD)	58	5–56	28.38	10.63
Similarities (SI)	46	2–41	21.82	6.90
Matrix Reasoning (MR)	32	3–28	18.77	4.20
Digit Span (DS)	54	4–38	24.48	6.45
Coding (CD)	117	13–97	47.22	18.29
Vocabulary (VC)	54	4–47	25.48	10.48
Figure Weights (FW)	34	3–31	21.76	4.80
Visual Puzzles (VP)	29	2–26	15.69	4.76
Picture Span (PS)	49	1–46	28.43	8.04
Symbol Search (SS)	60	3–60	28.28	9.32
Information (IN)	31	1–23	16.63	4.74
Picture Concepts (PC)	27	1–21	13.22	3.98
Letter-Number Sequencing (LN)	30	3–24	15.71	4.12
Cancellation (CA)	128	15–118	61.12	21.13
Comprehension (CO)	38	2–34	15.63	5.90
Arithmetic (AR)	34	4–29	17.77	5.71

Note: The results based on the total sample of 221 children aged 6–16.

**Table 2 jintelligence-13-00076-t002:** Item discrimination and difficulty for the WISC-V-ID subtest scores.

Subtest	Range of Score	Item Discrimination			Item Difficulty		
		Range	Mean	SD	Range	Mean	SD
Block Design (BD)	0–7	0.11–0.75	0.45	0.22	0.13–3.69	2.17	1.12
Similarities (SI)	0–2	0.25–0.68	0.43	0.13	0.05–1.93	0.94	0.70
Matrix Reasoning (MR)	0–1	−0.12–0.63	0.34	0.20	0.06–1.00	0.59	0.32
Digit Span (DS)	0–1	0.07–0.56	0.37	0.12	0.00–0.99	0.49	0.38
Coding (CD)	Not calculated						
Vocabulary (VC)	0–2	0.08–0.73	0.47	0.17	0.22–1.91	0.87	0.40
Figure Weights (FW)	0–1	−0.08–0.65	0.38	0.21	0.14–0.99	0.64	0.32
Visual Puzzles (VP)	0–1	0.02–0.62	0.39	0.18	0.06–0.99	0.54	0.33
Picture Span (PS)	0–2	0.11–0.6	0.44	0.12	0.01–1.94	1.09	0.63
Symbol Search (SS)	Not calculated						
Information (IN)	0–1	0.03–0.67	0.44	0.17	0.00–0.99	0.50	0.35
Picture Concepts (PC)	0–1	0.07–0.60	0.34	0.14	0.01–0.97	0.49	0.33
Letter-Number Sequencing (LN)	0–1	0.06–0.71	0.40	0.17	0.01–0.99	0.54	0.39
Cancellation (CA)	Not calculated						
Comprehension (CO)	0–2	0.14–0.63	0.45	0.14	0.01–1.95	0.82	0.59
Arithmetic (AR)	0–1	−0.01–0.72	0.40	0.18	0.01–1.00	0.52	0.33

Note: The results based on the total sample of 221 children aged 6–16.

**Table 3 jintelligence-13-00076-t003:** Cronbach’s α, McDonald’s ω, and standard error of measurement (SEM) by age group and overall sample following reordering and application of discontinue rules.

Subtest	*N* of Items	Cronbach’s α by Age Group				McDonald’s ω by Age Group			
		6–9 (*n* = 67)	10–12 (*n* = 66)	13–16 (*n* = 88)	Overall (*n* = 211)	6–9 (*n* = 67)	10–12 (*n* = 66)	13–16 (*n* = 88)	Overall (*n* = 211)
BD	13	**0.81** (4.01)	**0.78** (4.85)	**0.74** (4.74)	**0.81** (4.79)	**0.85** (3.56)	**0.84** (4.14)	**0.84** (3.72)	**0.87** (3.97)
SI	23	**0.88** (2.51)	**0.77** (2.44)	**0.80** (2.56)	**0.86** (2.63)	**0.90** (2.29)	**0.76** (2.49)	**0.81** (2.50)	**0.88** (2.51)
MR	32	**0.89** (1.60)	**0.85** (1.66)	**0.88** (1.22)	**0.88** (1.66)	**0.90** (1.52)	**0.86** (1.60)	**0.80** (1.57)	**0.89** (1.59)
DS	54	**0.90** (2.01)	**0.82** (1.94)	**0.80** (2.13)	**0.89** (2.11)	**0.91** (1.91)	**0.84** (1.83)	**0.80** (2.13)	**0.90** (2.06)
CD	Not Calculated								
VC	29	**0.84** (2.35)	**0.89** (7.63)	**0.86** (3.36)	**0.93** (3.16)	**0.85** (2.28)	**0.90** (7.28)	**0.88** (3.11)	**0.94** (2.89)
FW	34	**0.90** (1.57)	**0.91** (4.38)	**0.88** (1.53)	**0.91** (1.67)	**0.92** (1.40)	**0.93** (3.86)	**0.89** (1.47)	**0.93** (1.5)
VP	29	**0.85** (1.62)	**0.91** (3.84)	**0.88** (1.68)	**0.89** (1.71)	**0.86** (1.56)	**0.91** (3.84)	**0.90** (1.53)	**0.91** (1.59)
PS	26	**0.88** (2.84)	**0.84** (2.85)	**0.83** (2.90)	**0.88** (2.98)	**0.89** (2.72)	**0.86** (2.67)	**0.84** (2.81)	**0.90** (2.71)
SS	Not Calculated								
IN	31	**0.85** (1.40)	**0.85** (1.37)	**0.80** (1.54)	**0.90** (1.57)	**0.87** (1.30)	**0.86** (1.32)	**0.84** (1.37)	**0.91** (1.37)
PC	27	**0.85** (1.54)	**0.84** (1.62)	**0.79** (1.61)	**0.86** (1.62)	**0.86** (1.49)	**0.85** (1.57)	**0.80** (1.58)	**0.87** (1.55)
LN	30	**0.90** (1.55)	**0.83** (1.40)	**0.72** (1.46)	**0.87** (1.55)	**0.91** (1.47)	**0.82** (1.44)	**0.47** (2.02)	**0.88** (1.53)
CA	Not Calculated								
CO	19	**0.77** (2.00)	**0.72** (2.35)	**0.80** (2.46)	**0.85** (2.35)	**0.79** (1.91)	**0.70** (2.43)	**0.82** (2.33)	**0.85** (2.35)
AR	34	**0.90** (1.68)	**0.83** (1.84)	**0.82** (1.89)	**0.90** (1.88)	**0.89** (1.77)	**0.85** (1.73)	**0.84** (1.78)	**0.91** (1.79)

Note: Bold = internal consistency; regular = standard error of measurement. BD = Block Design; SI = Similarities; MR = Matrix Reasoning; DS = Digit Span; CD = Coding; VC = Vocabulary; FW = Figure Weights; VP = Visual Puzzles; PS = Picture Span; SS = Symbol Search; IN = Information; PC = Picture Concepts; LN = Letter–Number Sequencing; CA = Cancellation; CO = Comprehension; AR = Arithmetic.

**Table 4 jintelligence-13-00076-t004:** Goodness of fit statistic for confirmatory factor analysis.

Age Group	Goodness of Fit Index						
	χ^2^	df	CFI	TLI	RMSEA	SMSR	AIC
Model 1							
6–9	141.80	99	0.94	0.92	0.08	0.06	6239
10–12	142.31	99	0.92	0.90	0.08	0.08	6168
13–16	179.23	99	0.87	0.84	0.10	0.09	8216
All Ages	240.10	99	0.95	0.94	0.08	0.04	20,944
Model 2							
6–9	128.38	97	0.95	0.94	0.07	0.06	6230
10–12	130.72	97	0.93	0.92	0.07	0.08	6160
13–16	157.98	97	0.90	0.88	0.08	0.09	8198
All Ages	210.38	97	0.96	0.95	0.07	0.04	20,919

**Table 5 jintelligence-13-00076-t005:** Reliability estimates for WISC-V-ID composite scores.

	Age Group			
Composite	6–9	10–12	13–16	Overall
Verbal Comprehension (Gc)	0.90	0.86	0.88	0.93
Visual Spatial (Gv)	0.79	0.83	0.74	0.84
Fluid Reasoning (Gf)	0.73	0.65	0.69	0.80
Working Memory (Gwm)	0.90	0.82	0.78	0.91
Processing Speed (Gs)	0.79	0.81	0.59	0.89
Full Scale IQ	0.93	0.92	0.92	0.96

Note: Composite score based on model 1.

**Table 6 jintelligence-13-00076-t006:** Correlations between the WISC-V-ID subtest scores and age.

Subtest	BD	SI	MR	DS	CD	VC	FW	VP	PS	SS	IN	PC	LN	CA	CO	AR
Age *	0.53	0.61	0.55	0.65	0.76	0.71	0.46	0.44	0.53	0.58	0.72	0.53	0.64	0.72	0.68	0.67

Note: * All correlations were statistically significant (*p* < 0.001). Note: Bold = internal consistency; regular = standard error of measurement. BD = Block Design; SI = Similarities; MR = Matrix Reasoning; DS = Digit Span; CD = Coding; VC = Vocabulary; FW = Figure Weights; VP = Visual Puzzles; PS = Picture Span; SS = Symbol Search; IN = Information; PC = Picture Concepts; LN = Letter–Number Sequencing; CA = Cancellation; CO = Comprehension; AR = Arithmetic.

## Data Availability

The informed consent forms did not specifically ask for permission to store and share the data in a public repository. However, the fully anonymized data are available from the corresponding author upon reasonable request.

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
