# Peer review of "The Adaptation of the Wechsler Intelligence Scale for Children—5th Edition (WISC-V) for Indonesia: A Pilot Study"

_jintelligence, 2025, doi:10.3390/jintelligence13070076_

Round 1
Reviewer 1 Report
Comments and Suggestions for Authors
This article presents the first results of the Indonesian adaptation of the WISC-V (translation and item try-out). It is a simple technical report, with no methodological innovation or results of interest to scientific knowledge.
Moreover, several points of the methodology are questionable:
- The authors emphasize their use of backtranslation, even though this procedure is criticized in the standards of the International Test Commission (standard TD2-(5)).
- Item scaling was not rigorously controlled. For this purpose Item Response Theory should be used.
- Although Cronbach's alpha was used in the original 2014 version of the WISC-V, McDonald's Omega coefficient is now recommended.
- Considering that an alpha between 0.60 and 0.70 is acceptable is highly debatable. The authors refer to Kline (1998), but many others are much more demanding. Thus, Nunally (1994) considers that "if important decisions are made with respect to specific test scores, a reliability of .90 is the bare minimum, and a reliability of .95 should be considered the desirable standard."
- The authors refer to the article by von Davier et al. (2019) to correct their reliability coefficients after revision of the item order and the application of discontinue rules. The conclusion of von Davier et al. (p.162) was: “The recommended approach is not to code the non-observed data as wrong, but rather to retain the missing data status for those items that were eliminated from administration by applying a discontinue rule." This is not what the authors of the present article did, since their participants responded to all the items. In this case, there is no need to correct the reliability coefficients, which must be calculated from the observed scores.
Author Response
Thank you very much for taking the time to review this manuscript. While this manuscript may appear as a technical report without novel methods or findings, its value lies in providing a much-needed adaptation of a widely used test for the Asian context. Most adaptations have been conducted in Europe, so this work helps fill an important regional gap and supports researchers and practitioners in Asia. Please find the detailed responses below and the corresponding highlighted in the re-submitted files.

Reviewer 2 Report
Comments and Suggestions for Authors
The authors present a fully translated and culturally adapted Indonesian version of the Wechsler Intelligence Scale for Children – Fifth Edition (WISC-V). Using complete administration data from 221 children aged 6–16 years, they compare reliability under two conditions—original item order + no discontinue rule versus re-ordered item order + discontinue rule—and propose an integrated improvement package. The paper also reports item difficulty, item–total correlations, and internal‐consistency coefficients for each subtest. Overall, the study’s logic is clear, and the data analyses and conclusions appear sound.
Issues for Consideration
- Research Question 3 (“reorder the item sequence of the subtests according to the empirical item difficulty observed in Indonesian children's responses”)
Only two conditions—original + no discontinue and re-ordered + discontinue—are reported, so the independent effects of item order and discontinue rule cannot be disentangled. Please add the remaining two conditions (original + discontinue; re-ordered + no discontinue) with their α/SEM values, or run a 2 × 2 factorial analysis to clarify main and interaction effects.
- Internal IRT Calibration
Conduct a 1-PL/Rasch calibration (2-PL if sample size allows) for all items and report item fit indices and the Test Information Function. This will verify:
(i) whether the empirically re-ordered difficulty hierarchy matches IRT estimates;
(ii) whether the discontinue rule reduces information at the high or low ends of the θ continuum.
- Negative-discrimination items
Eight items show negative discrimination, yet are retained without quantitative justification. Please provide at least one of:
Cronbach’s α after removing these items, or
Rasch infit/outfit statistics for them,
and explain statistically why they are kept.
- Future cross-cultural structural validity
In the Discussion, outline plans for CHC-based CFA/ESEM and multi-group invariance testing to address cross-cultural generalizability of the WISC-V factor structure.
- Quantifying cultural bias
The manuscript notes that non-local norms may distort ability estimates, but relies only on translation differences or shifts in item-difficulty rank as indirect evidence. If feasible, at least a indirect DIF check would give more explicit quantitative evidence of cultural bias.
Author Response
Thank you very much for taking the time to review this manuscript. I also thank for your positive impression of our manuscript. Please find the detailed responses below and the corresponding highlighted in the re-submitted files.

Reviewer 3 Report
Comments and Suggestions for Authors
The authors are to be commended for performing a valuable service for Indonesian assessment. They are also to be commended for following excellent procedures for adapting and translating the WISC-V to Indonesian language and culture. However, their data are not interpretable as currently analyzed. The age range is too heterogeneous to permit valid interpretation of item analysis or reliability data. All obtained coefficients are spuriously inflated by the wide age range. In addition, they should have conducted a small test-retest study to be able to provide reliability estimates for the Processing Speed subtests. I believe the manuscript will be publishable if the authors do the following (even if they are unable to retest any subjects on the speed subtests:
1. Divide the sample into three portions of about 70 individuals each. Ideally something like ages 6-8, 9-12, and 13-16 will result in samples of about 60-80 individuals each.
2. Redo all analyses for the three samples that are more homogeneous on the variable of age.
3. Conduct factor analysis, both exploratory and confirmatory for the three age groups. Alternatively (or in addition) the authors can conduct these factor analyses on the total sample if they use a partial correlation matrix (removing the influence of chronological age). The WISC-V is interpreted as a measure of 5 cognitive abilities + Full Scale IQ. Its construct validity must be examined in this preliminary study. Once the authors conduct these analyses, they must compute the reliability of each meaningful factor they obtain, which including “g”.
4. The authors must cite current research on the American WISC-V, including the two most popular books on the test—Flanagan and Alfonso’s Essentials of WISC-V Assessment and Kaufman et al’s Intelligent Testing with the WISC-V, both published by John Wiley.
Author Response
Thank you very much for your kind words regarding our study. We agree that this research is important for advancing psychological assessment in Indonesia and encouraging high-quality, standardized adaptation efforts.
Your remark about conducting the analysis based on age groups is highly valuable. Accordingly, we have recalculated the analyses following your suggestion. The updated results can be found in the Method, Results, and Discussion sections. We hope these revisions address your concern.
Due to time constraints, test-retest analysis was not conducted in this pilot study. However, we plan to do this research and report it in a future publication. Please find the detailed responses below and the corresponding highlighted in the re-submitted files.

Round 2
Reviewer 1 Report
Comments and Suggestions for Authors
The authors have taken into account the comments made on the first version of their paper. This new version is a significant improvement on the previous one. The analyses and discussion of score reliability and factor structure are appropriate and interesting. This article is not very original, but provides useful information in the debate on the factor structure of the WISC-V. It can be published in its current version.
Author Response
We thank the reviewer for the positive and constructive remarks, as well as for the recommendation to publish the manuscript. We hope this article will contribute useful information to the ongoing debate on the factor structure of the WISC-V.
Reviewer 3 Report
Comments and Suggestions for Authors
The authors are to be commended for conducting the additional analyses I recommended. The results of these analyses greatly enhance the value of the article. The authors should delete Table 3. It is redundant with Table 4. It is sensible for the authors to reorder the items based on difficulty levels for Indonesian children and to apply the same discontinue rules that are used in the American version. That is the version that will be used in clinical practice and that is the only version for which reliability estimates should be reported. The values in Table 4 are pertinent. The values in Table 3 are not and they flood the article with unnecessary data. Similarly, the authors should delete their 2 X 2 analyses involving the two sets of coefficients. They do not provide important information. Coefficient alpha is not affected by item order. And of course applying the discontinue rules will lead to lower raw scores, on average, because points earned on items beyond the discontinue rule are subtracted from a person’s raw score. This analysis weakens, not strengthens, the article. In the Discussion the authors note that the reliability increases for some subtests when some “bad” items are deleted. That will almost always occur, but whether or not such an increase will actually be observed is if a totally new sample is tested. You can’t draw any conclusions about the existing “computation” sample; there must be a cross-validation sample as well.
Author Response
We thank the reviewer for these positive remarks. We have addressed the remaining issues in our rebuttal and in the highlighted second revision of our manuscript.

Round 3
Reviewer 3 Report
Comments and Suggestions for Authors
I fully support the acceptance of the manuscript in its current form.